# The Adversarial Regulation of the Temporal Difference Loss Costs More Than Expected

## Abstract

Deep reinforcement learning research has enabled reaching significant performance levels for sequential decision making in MDPs with highly complex observations and state dynamics with the aid of deep neural networks. However, this aid came with a cost that is inherent to deep neural networks which have increased sensitivities towards indistinguishable peculiarly crafted non-robust directions. To alleviate these sensitivities several studies suggested techniques to cope with this problem via explicitly regulating the temporal difference loss for the worst-case sensitivity. In our study, we show that these worst-case regularization techniques come with a cost that intriguingly causes inconsistencies and overestimations in the state-action value functions. Furthermore, our results essentially demonstrate that vanilla trained deep reinforcement learning policies have more accurate and consistent estimates for the state-action values. We believe our results reveal foundational intrinsic properties of the adversarial training techniques and demonstrate the need to rethink the approach to robustness in deep reinforcement learning.

## 1 Introduction

Advancements in deep neural networks have recently proliferated leading to expansion in the domains where deep neural networks are utilized including image classification (Krizhevsky et al., 2012), natural language processing (Sutskever et al., 2014), speech recognition (Hannun et al., 2014) and self learning systems via exploration. In particular, deep reinforcement learning has become an emerging field with the introduction of deep neural networks as function approximators (Mnih et al., 2015). Hence, deep neural policies have been deployed in many different domains from pharmaceuticals to self driving cars (Daochang & Jiang, 2018; Huan-Hsin et al., 2017; Noonan, 2017).

As the advancements in deep neural networks continued a line of research focused on their vulnerabilities towards a certain type of specifically crafted perturbations computed via the cost function used to train the neural network (Szegedy et al., 2014; Goodfellow et al., 2015; Madry et al., 2018; Kurakin et al., 2016; Dong et al., 2018). While some research focused on producing optimal $\ell_p$-norm bounded perturbations to cause the most possible damage to the deep neural network models, an extensive amount of work focused on making the networks robust to such perturbations (Madry et al., 2018; Carmon et al., 2019; Raghunathan et al., 2020).

The vulnerability to such particularly optimized adversarial directions was inherited by deep neural policies as well (Huang et al., 2017; Kos & Song, 2017; Korkmaz, 2022). Thus, robustness to such perturbations in deep reinforcement learning became a concern for the machine learning community, and several studies proposed various methods to increase robustness (Pinto et al., 2017; Gleave et al., 2020). Thus, in this paper we focus on adversarially trained deep neural policies and the state-action value function learned by these training methods in the presence of an adversary. In more detail, in this paper we aim to seek answers for the following questions: (i) How accurate is the state-action value function on estimating the values for state-action pairs in MDPs with high dimensional state representations?, (ii) Does adversarial training affect the estimates of the state-action value function?, (iii) What are the effects of training with worst-case distributional shift on the state-action value function representation for the optimal actions? and (iv) Are there any fundamental trade-offs intrinsic to explicit worst-case regularization in deep neural policy training? To be able to answer these questions we focus on adversarial training and robustness in deep neural policies and make the following contributions:

- We conduct an investigation on the state-action values learnt by the state-of-the-art adversarially trained deep neural policies and vanilla trained deep neural policies.

- We provide theoretically motivated justification for how adversarial training might change the state-action value function.

- We perform several experiments in Atari games with large state spaces from the Arcade Learning Environment (ALE). With our systematic analysis we show that vanilla trained deep neural policies have a more accurate representation of the sub-optimal actions compared to the state-of-the-art adversarially trained deep neural policies.

- Furthermore, we show the inconsistencies in the action ranking in the state-of-the-art adversarially trained deep neural policies. Thus, these results demonstrate the loss of information in state-action value function as a novel fundamental trade-off intrinsic to adversarial training.

- More importantly, we demonstrate that state-of-the-art adversarially trained deep neural policies learn overestimated state-action value functions.

- Finally, we explain how our results call into question the hypothesis initially proposed by Bellemare et al. (2016) relating the action gap and overestimation.

## 2 BACKGROUND AND PRELIMINARIES

**Preliminaries:** In deep reinforcement learning the goal is to learn a policy for taking actions in a Markov Decision Process (MDP) that maximize discounted expected cumulative reward. An MDP is represented by a tuple $\mathcal{M} = (S, A, P, r, \rho_0, \gamma)$ where $S$ is a set of continuous states, $A$ is a discrete set of actions, $P$ is a transition probability distribution on $S \times A \times S$, $r : S \times A \to \mathbb{R}$ is a reward function, $\rho_0$ is the initial state distribution, and $\gamma$ is the discount factor. The goal in reinforcement learning is to learn a policy $\pi : S \to \mathcal{P}(A)$ which maps states to probability distributions on actions in order to maximize the expected cumulative reward $R = \mathbb{E} \sum_{t=0}^{T-1} \gamma^t r(s_t, a_t)$ where $a_t \sim \pi(s_t)$. In $Q$-learning Watkins (1989) the goal is to learn the optimal state-action value function $Q^*(s, a) = R(s, a) + \sum_{s' \in S} P(s'|s, a) \max_{a' \in A} Q^*(s', a')$. Thus, the optimal policy is determined by choosing the action $a^*(s) = \arg\max_a Q(s, a)$ in state $s$.

**Adversarial Crafting and Training:** Szegedy et al. (2014) observed that imperceptible perturbations could change the decision of a deep neural network and proposed a box constrained optimization method to produce such perturbations. Goodfellow et al. (2015) suggested a faster method to produce such perturbations based on the linearization of the cost function used in training the network. Kurakin et al. (2016) proposed the iterative version of the fast gradient sign method proposed by Goodfellow et al. (2015) inside an $\epsilon$-ball.

$$x_{\text{adv}}^{N+1} = \text{clip}_\epsilon(x_{\text{adv}}^N + \alpha \text{sign}(\nabla_x J(x_{\text{adv}}^N, y))) \tag{1}$$

in which $J(x, y)$ represents the cost function used to train the deep neural network, $x$ represents the input, and $y$ represents the output labels. While several other methods have been proposed (e.g. Korkmaz (2020)) using a momentum-based extension of the iterative fast gradient sign method,

$$v_{t+1} = \mu \cdot v_t + \frac{\nabla_{s_{\text{adv}}} J(s_{\text{adv}}^t + \mu \cdot v_t, a)}{\|\nabla_{s_{\text{adv}}} J(s_{\text{adv}}^t + \mu \cdot v_t, a)\|_1} \tag{2}$$

$$s_{\text{adv}}^{t+1} = s_{\text{adv}}^t + \alpha \cdot \frac{v_{t+1}}{\|v_{t+1}\|_2} \tag{3}$$

adversarial training has mostly been conducted with perturbations computed by projected gradient descent (PGD) proposed by Madry et al. (2018) (i.e. Equation 1).

**Adversaries and Training in Deep Neural Policies:** The initial investigation on resilience of deep neural policies was conducted by Kos & Song (2017) and Huang et al. (2017) concurrently based on the utilization of the fast gradient sign method proposed by Goodfellow et al. (2015). Korkmaz (2022) showed that deep reinforcement learning policies learn shared adversarial features across MDPs. While several studies focused on improving optimization techniques to compute optimal perturbations, a line of research focused on making deep neural policies resilient to these perturbations. Mandlekar et al. (2017) proposed including these perturbations in training time to increase resilience for robotic

setups. Pinto et al. (2017) proposed to model the dynamics between the adversary and the deep neural policy as a zero-sum game where the goal of the adversary is to minimize expected cumulative rewards of the deep neural policy. Gleave et al. (2020) approached this problem with an adversary model which is restricted to take natural actions in the MDP instead of modifying the observations with $\ell_p$-norm bounded perturbations. The authors model this dynamic as a zero-sum Markov game and solve it via self play. Recently, Huan et al. (2020) proposed to model this interaction between the adversary and the deep neural policy as a state-adversarial MDP, and claimed that their proposed algorithm State Adversarial Double Deep Q-Network (SA-DDQN) learns theoretically certified robust policies against natural noise and perturbations. More recently, several empirical concerns have been raised on the robustness of theoretically certified adversarially trained deep neural policies Ezgi (2021). In our work, we systematically investigate and theoretically motivate the problems caused by adversarial training on the state-action value function learned by deep neural policies.

## 3 ADVERSARIAL TRAINING AND THE STATE-ACTION VALUE FUNCTION

In this paper we aim to answer the following questions:

- *How does training with explicit worst-case regularization affect the estimates of the optimal state-action values in MDPs with high dimensional state representations?*
- *What is the accuracy of the state-action value function representation for the non-optimal actions in deep neural policies?*
- *Does state-of-the-art adversarial training affect the state-action value estimates?*
- *Are there any intrinsic trade-offs tied to adversarial deep neural policy training?*

While the goal in $Q$-learning is to learn the state-action value function $Q(s, a)$ that maximizes expected discounted cumulative rewards, in deep $Q$-learning an additional concern arises from susceptibility towards adversarial perturbations due to the nonlinear function approximator used in learning the $Q$-function. Ideally, one might hope that adversarial training would reduce the vulnerability of the $Q$-function to adversarial perturbations while preserving the $Q$-values of the non-perturbed states as much as possible. The theoretically motivated adversarial training techniques achieve certified defense against adversarial perturbations inside the $\epsilon$-ball $D_\epsilon(s) = \{\bar{s} : \|s - \bar{s}\|_\infty \leq \epsilon\}$. However, we show that this approach induces significant changes in the $Q$-function so that the $Q$-function loses its *accuracy* for the non-perturbed states. In particular, adversarial training causes deep neural policies to learn overestimated state-action values, and the $Q$-values for non-optimal actions are reduced in accuracy to the point where their relative ranking changes.

In the remainder of this section we give theoretical motivation for these empirical results. In particular we demonstrate that in the setting of linear function approximation, adversarial training can potentially lead to overestimation for the $Q$-values of the optimal actions, and reordering of the ranking of non-optimal actions. The basic approach of adversarial training techniques is based on adding a regularizer to the standard $Q$-learning update. The regularizer is designed to penalize $Q$-functions for which a perturbed state $\bar{s} \in D_\epsilon(s)$ can change the identity of the highest $Q$-value action. For the baseline adversarial training technique we will theoretically analyze the effects of this regularizer.

**Definition 3.1** (Huan et al. (2020)). For a state $s$ let $a^*(s) = \arg\max_a Q(s, a)$. The regularizer is given by

$$\mathcal{R}(\theta) = \sum_s \left( \max_{\bar{s} \in D_\epsilon(s)} \max_{a \neq a^*(s)} Q_\theta(\bar{s}, a) - Q_\theta(\bar{s}, a^*(s)) \right).$$

The adversarial training algorithm proceeds by adding $\mathcal{R}(\theta)$ to the standard temporal difference loss

$$\mathcal{L}(\theta) = L_H \left( r + \gamma \max_{a'} Q^{\text{target}}(s', a') - Q_\theta(s, a) \right) + \mathcal{R}(\theta) \tag{4}$$

used in DQN and minimizing via stochastic gradient descent.

We now describe the construction of an MDP $\mathcal{M}$ with linear function approximation where the use of the regularizer causes overestimation and reordering of suboptimal actions. There are two states parametrized by feature vectors $s_1, s_2 \in \mathbb{R}^n$, and there are three possible actions $\{a_i\}_{i=1}^3$ in each state. Taking any of the three actions in state $s_1$ leads to a transition to state $s_2$ and vice

versa. Let $1 > \gamma > 0$ be the discount factor, and let $\delta > \eta > 0$ be small constants with $\gamma > \delta$. The rewards for each action are as follows: $r(s_1, a_1) = 1 - \gamma$, $r(s_1, a_2) = \eta - \gamma$, $r(s_1, a_3) = \delta - \gamma$, $r(s_2, a_1) = \eta - \gamma$, $r(s_2, a_2) = 1 - \gamma$, and $r(s_2, a_3) = \delta - \gamma$. Clearly, the optimal policy is to always take action $a_1$ in state $s_1$, and action $a_2$ in state $s_2$ as these are the only actions giving positive reward. Thus the optimal state-action values are given by: $Q^*(s_1, a_1) = Q^*(s_2, a_2) = \sum_{t=0}^{\infty}(1 - \gamma)\gamma^t = 1$, $Q^*(s_1, a_2) = Q^*(s_2, a_1) = \eta - \gamma + \gamma\sum_{t=0}^{\infty}(1 - \gamma)\gamma^t = \eta$, and $Q^*(s_1, a_3) = Q^*(s_2, a_3) = \delta - \gamma + \gamma\sum_{t=0}^{\infty}(1 - \gamma)\gamma^t = \delta$. Let the $Q$-function be linearly parametrized by $\theta = (\theta_1, \theta_2, \theta_3)$ so that $Q_\theta(s, a_i) = \langle\theta_i, s\rangle$. Finally, let $z_i$ for $i \in \{1, 2, 3\}$ be three orthonormal vectors, and let the state feature vectors satisfy:

$$1.\ s_1 = z_1 + \delta z_3 + \eta z_2 \quad \text{and} \quad 2.\ s_2 = z_2 + \delta z_3 + \eta z_1$$

Then it follows that the optimal $Q$-function is parametrized by $\theta^* = (\theta_1^*, \theta_2^*, \theta_3^*)$ where $\theta_i^* = z_i$ i.e. $Q_{\theta^*}(s, a) = Q^*(s, a)$ for all $s$ and $a$. Thus, according to the function $Q_{\theta^*}(s, a)$, for $s_1$ the best action is $a_1$, for $s_2$ the best action is $a_2$, and in all states the second-best action is $a_3$. Next we identify the optimal perturbations used in the computation of the regularizer $\mathcal{R}(\theta^*)$ for this setting.

**Proposition 3.1.** *In the MDP $\mathcal{M}$ suppose that $\epsilon < \frac{\delta - \eta}{2}$.*

1. *For $s = s_1$: $s + \frac{\epsilon}{\sqrt{2}}(\theta_3^* - \theta_1^*) = \arg\max_{\bar{s} \in D_\epsilon(s)} \max_{a \neq a^*(s)} Q_{\theta^*}(\bar{s}, a) - Q_{\theta^*}(\bar{s}, a^*(s))$*

2. *For $s = s_2$: $s + \frac{\epsilon}{\sqrt{2}}(\theta_3^* - \theta_2^*) = \arg\max_{\bar{s} \in D_\epsilon(s)} \max_{a \neq a^*(s)} Q_{\theta^*}(\bar{s}, a) - Q_{\theta^*}(\bar{s}, a^*(s))$*

*Proof.* We will prove item 1, and item 2 will follow from an identical argument with roles of $\theta_1^*$ and $\theta_2^*$ swapped. Let $s = s_1$. Any $\bar{s} \in D_\epsilon(s)$ can be written as $s + \epsilon v$ where $v$ is a unit vector. Thus, $\langle\theta_3^*, \bar{s}\rangle = \langle\theta_3^*, s\rangle + \epsilon\langle\theta_3^*, v\rangle > \langle\theta_3^*, s\rangle - \epsilon = \delta - \epsilon$. Similarly we have $\langle\theta_2^*, \bar{s}\rangle < \langle\theta_2^*, s\rangle + \epsilon = \eta + \epsilon$. Since $\epsilon < \frac{\delta - \eta}{2}$, we conclude that $\langle\theta_3^*, \bar{s}\rangle > \langle\theta_2^*, \bar{s}\rangle$ for all $\bar{s} \in D_\epsilon(s)$. Therefore, in state $s$ the action maximizing $\max_{a \neq a^*(s)} Q_{\theta^*}(\bar{s}, a) - Q_{\theta^*}(\bar{s}, a^*(s))$ will always be $a_3$. This implies that

$$\arg\max_{\bar{s} \in D_\epsilon(s)} \max_{a \neq a^*(s)} Q_{\theta^*}(\bar{s}, a) - Q_{\theta^*}(\bar{s}, a^*(s)) = \arg\max_{\bar{s} \in D_\epsilon(s)}\langle\theta_3^*, \bar{s}\rangle - \langle\theta_1^*, \bar{s}\rangle. \tag{5}$$

This is the maximum in a ball of radius $\epsilon$ around $s$ of the linear function $\langle\theta_3^* - \theta_1^*, \bar{s}\rangle$. Therefore the maximum is achieved by $\bar{s} = s + \frac{\epsilon}{\sqrt{2}}(\theta_3^* - \theta_1^*)$ as desired. $\square$

In words, the optimal direction to perturb the state $s_1$ in order to have $a^*(s) \neq a^*(\bar{s})$ is toward $\theta_3^* - \theta_1^*$. Similarly for the state $s_2$, the optimal perturbation is toward $\theta_3^* - \theta_2^*$. Next we use this fact to show that in order to decrease the regularizer it is sufficient to simply increase the magnitude of $\theta_1$ and $\theta_2$, and decrease the magnitude of $\theta_3$.

**Proposition 3.2.** *In the MDP $\mathcal{M}$ let $\lambda > 0$ and suppose that $\epsilon < \frac{(1-\lambda)\delta - (1+\lambda)\eta}{2}$. Let $\theta = (\theta_1, \theta_2, \theta_3)$ be given by $\theta_1 = (1 + \lambda)\theta_1^*$, $\theta_2 = (1 + \lambda)\theta_2^*$ and $\theta_3 = (1 - \lambda)\theta_3^*$. Then $\mathcal{R}(\theta) < \mathcal{R}(\theta^*)$.*

*Proof.* By an identical argument to that in Proposition 3.1 we have that $a_3$ is always the action maximizing $\max_{a \neq a^*(s)} Q_\theta(\bar{s}, a) - Q_\theta(\bar{s}, a^*(s))$ whenever $\epsilon < \frac{(1-\lambda)\delta - (1+\lambda)\eta}{2}$. This condition is satisfied by assumption. Therefore, we conclude that for $s = s_1$, the optimal $\bar{s} \in D_\epsilon(s)$ for the scaled parameters $\theta$ is given by $\bar{s} = s + \frac{\epsilon}{\sqrt{2(1+\lambda^2)}}(\theta_3 - \theta_1)$. Therefore, the contribution to the sum defining $\mathcal{R}(\theta)$ from state $s_1$ is given by

$$\langle(\theta_3 - \theta_1), \bar{s}\rangle = \langle(\theta_3 - \theta_1), s\rangle + \epsilon\sqrt{2(1+\lambda^2)} = -(1+\lambda) + (1-\lambda)\delta + \epsilon\sqrt{2(1+\lambda^2)} \tag{6}$$

where the last step uses the fact that $s = \theta_1^* + \delta\theta_3^* + \eta\theta_2^*$ and that the vectors $\theta_i^*$ are orthonormal. Next using the fact that $\sqrt{1 + \lambda^2} < 1 + \lambda$ for all $\lambda > 0$ we conclude

$$\langle(\theta_3 - \theta_1), \bar{s}\rangle < -(1+\lambda) + (1-\lambda)\delta + \epsilon\sqrt{2} + \epsilon\lambda\sqrt{2} < -(1+\lambda) + \delta + \epsilon\sqrt{2}. \tag{7}$$

The final inequality follows from the fact that $\epsilon < \frac{\delta}{2}$ so $\epsilon\lambda\sqrt{2} - \lambda\delta < 0$. Switching to type 2 actions an identical proof (with $\theta_1$ replaced by $\theta_2$) yields the same value for the contribution of type 2 actions to the sum. By Proposition 3.1, the contribution of each type of state to the sum defining $\mathcal{R}(\theta^*)$ is

$$\langle(\theta_3^* - \theta_1^*), s + \frac{\epsilon}{\sqrt{2}}(\theta_3^* - \theta_1^*)\rangle = -1 + \delta + \epsilon\sqrt{2}. \tag{8}$$

Clearly the contribution of each state in 7 is strictly less than that in 8. Therefore $\mathcal{R}(\theta) < \mathcal{R}(\theta^*)$. $\square$

**Theorem 3.3.** *If a linear state-action value function approximator $Q_\theta(s, a)$ is used, then the regularizer $\mathcal{R}(\theta)$ can lead to overestimation of the value of the optimal action, and re-ordering of the values of the suboptimal actions.*

*Proof.* Consider the MDP $\mathcal{M}$ with linear function approximation constructed above. Increasing the magnitude of $\theta_1^*$ and $\theta_2^*$ by a factor of $1 + \lambda$ leads to overestimation of the $Q$-value of the best action in both state $s_1$ and $s_2$ by the same factor. Additionally decreasing the magnitude of $\theta_3^*$ can lead to a change in the ranking of the suboptimal actions. Indeed if $\frac{1+\lambda}{1-\lambda} > \frac{\delta}{\eta}$ then $a_3$ will become the third ranked action in both states. Therefore, Proposition 3.2 proves that changing $\theta$ to decrease the regularizer $\mathcal{R}(\theta)$ can lead to both overestimation of the first ranked action, and re-ordering of the ranking of the suboptimal actions. □

While we showed how this can potentially happen in the case of linear function approximation, we will see that this is a general phenomenon which occurs with neural-network approximation of the $Q$-function in adversarially trained agents. It is important to note that the issues we identify are a result of the fundamental differences between deep neural policies and classification tasks where adversarial training has previously been applied. In particular, the fact that the state-action value function $Q(s, a)$ has a meaning (i.e. measuring expected cumulative rewards) with regard to the MDP beyond simply labelling the optimal action correctly is the root cause of the effects that we observe. In other words, simply penalizing the state-action value function for assigning the wrong "label" to an adversarial example can have unintended, potentially detrimental consequences for learning an accurate state-action value function.

## 4    MEASURING THE ACCURACY OF STATE-ACTION VALUES

In this section we provide a methodology to measure the accuracy of the state-action value function in representing values for the non-optimal actions. At a high-level, our approach is based on action modification and the relative performance drop $\mathcal{P}$ as defined below:

**Definition 4.1.** The performance drop of an agent when modifying the agent's actions is given by

$$\mathcal{P} = \frac{\text{Score}_{\text{base}} - \text{Score}_{\text{actmod}}}{\text{Score}_{\text{base}} - \text{Score}_{\text{min}}}. \tag{9}$$

where $\text{Score}_{\text{base}}$ represent the baseline run of the game with no action modification, $\text{Score}_{\text{min}}$ represents the minimum score available for a given game, and $\text{Score}_{\text{actmod}}$ represents the run of the game where the actions of the agent are modified for a fraction of the state observations.

We now explain precisely how we propose to measure "accuracy" for non-optimal actions. Formally, let $a_i$ be the $i^{\text{th}}$ best action decided by the deep neural policy in a given state $s$ (i.e. $Q(s, a)$ is sorted in decreasing order, and $a_i$ is the action corresponding to $i^{\text{th}}$ largest $Q$-value). For a trained agent, the value of $Q(s, a_i)$ should represent the expected cumulative rewards obtained by taking action $a_i$ in state $s$, and then taking the highest $Q$-value action (i.e. $a_1$) in every subsequent state. Thus, a natural test to perform would be: pick a random state $s$, make the agent choose action $a_i$ in state $s$, and in all other states have the agent choose the highest $Q$-value action. By comparing the relative performance drop $\mathcal{P}$ in this test to a clean run where the agent always takes the highest $Q$-value action, one can measure the decline in rewards caused by taking action $a_i$. Further, we can provide a measure of accuracy for the state-action value function by comparing the results of the test for each $i \in \{1, 2 \ldots |A|\}$, and checking that the relative performance drops $\mathcal{P}_i$ are in the correct order i.e. $0 = \mathcal{P}_1 \leq \mathcal{P}_2 \cdots \leq \mathcal{P}_{|A|}$.

However, there is an issue with the above proposal. It is often the case that there are many states $s$ in which the action taken has very little impact on the final rewards. Instead, there are a relatively smaller number of critical states in which the action taken has a large impact. Thus, picking a single random state $s$ in which to take action $a_i$ will have a statistically insignificant impact on the final rewards in the game. Therefore we modify the test described above by instead sampling a $p$-fraction of the states in the episode uniformly at random, and making the agent take action $a_i$ in each of the sampled states. We then record the relative performance drop as a function of $p$, yielding a reward curve $\mathcal{P}_i(p)$. More formally, we define

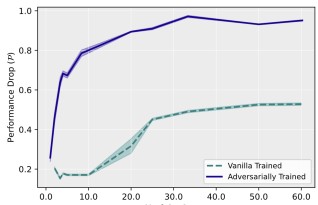 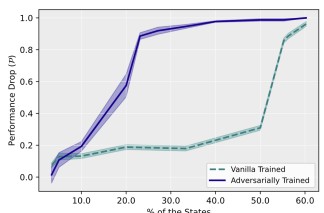 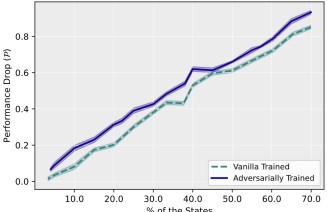

Figure 1: Performance drop $\mathcal{P}_2(p)$ with respect to action modification $a_w$ for the state-of-the-art adversarially trained deep neural policies and vanilla trained deep neural policies. Left: BankHeist. Center: RoadRunner. Right: Freeway.

**Definition 4.2.** Let $\mathcal{M}$ be an MDP and $Q(s,a)$ be a state-action value function for $\mathcal{M}$. In each state label the actions $a_1, \ldots a_{|A|}$ in order so that $Q(s, a_1) \geq Q(s, a_2) \cdots \geq Q(s, a_{|A|})$. We define the *performance curve* $\mathcal{P}_i(p)$ to be the expected performance drop of an agent in $\mathcal{M}$ which takes action $a_i$ in a randomly sampled $p$-fraction of states, and takes action $a_1$ in all other states.

Using these reward curves one can check whether $\mathcal{P}_i(p)$ lies above $\mathcal{P}_j(p)$ whenever $i > j$. Of course one curve may not always lie strictly above or below another, so we introduce the following definition to quantitatively capture the relative ordering of performance drop curves.

**Definition 4.3.** Let $F : [0,1] \to [0,1]$ and $G : [0,1] \to [0,1]$. For any $\tau > 0$, we say that the $F$ $\tau$-dominates $G$ if $\int_0^1 (F(p) - G(p))\, dp > \tau$.

To compare the accuracy of state-action values for vanilla versus adversarially trained agents, we can thus perform the above test, and check the relative ordering of the curves $\mathcal{P}_i(p)$ using Definition 4.3 for each agent type. In addition, we can also directly compare for each $i$ the curve $\mathcal{P}_i^{\mathrm{adv}}(p)$ for the adversarially trained agent with the curve $\mathcal{P}_i^{\mathrm{vanilla}}(p)$ for the vanilla trained agent. This is possible because $\mathcal{P}_i(p)$ measures the performance drop of the agent relative to a clean run, and so always takes values on a normalized scale from 0 to 1. Thus, if we observe for example that $\mathcal{P}_2^{\mathrm{adv}}(p)$ $\tau$-dominates $\mathcal{P}_2^{\mathrm{vanilla}}(p)$ for some $\tau > 0$, we can conclude that the state-action value function of the vanilla trained agent more accurately represents the second-best action than that of the adversarially trained agent.

## 5 EXPERIMENTAL DETAILS

The experiments are conducted in high dimensional state representation MDPs. In particular, our experiments are conducted in the Arcade Learning Environment (ALE) (Bellemare et al., 2013) in the OpenAI (Brockman et al., 2016) baseline version. The vanilla trained deep neural policy is trained via Double Deep Q-Network (DDQN) (Wang et al., 2016) initially proposed by Hasselt et al. (2016) with prioritized experience replay proposed by (Schaul et al., 2016), and the state-of-the-art adversarially trained deep neural policy is trained via State-Adversarial Double Deep Q-Network (SA-DDQN) (Section 2) with prioritized experience replay (Schaul et al., 2016). The results are averaged over 10 episodes. We explain in detail all the necessary hyperparameters for the implementation in the supplementary material. The standard error of the mean is included for all of the figures and tables. Note that in the main body of the paper we focus on the baseline adversarial training. In the supplementary material we also provide analysis on the follow-up more recent studies in adversarial training techniques. The results reported for all of the adversarial training techniques remains the same that the adversarially trained policies learn inaccurate, inconsistent and overestimated state-action values.

## 6 AN ANALYSIS ON THE STATE-ACTION VALUE FUNCTION REPRESENTATION

In this section we demonstrate that the state-action value function of adversarially trained deep neural policies provides inaccurate estimates for the non-optimal actions, and learns overestimated state-action values. This confirms that the theoretically-motivated problems discussed in Section 3 do indeed occur in practice for deep neural policies. In particular, to evaluate the accuracy on non-optimal actions we use the methodology discussed in Section 4 of measuring the performance drop $\mathcal{P}_i(p)$ that occurs when causing the deep neural policy to take the $i$-th best action in a $p$ fraction

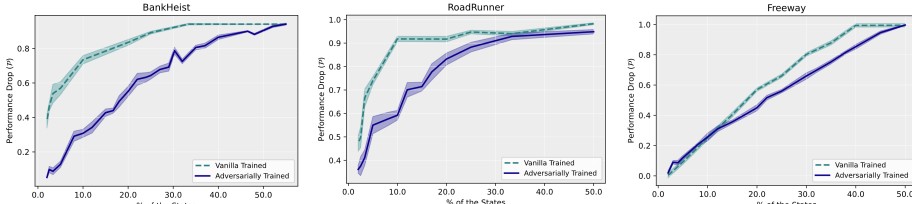

Figure 2: Performance drop $\mathcal{P}_w(p)$ with respect to action modification $a_w$ for the state-of-the-art adversarially trained deep neural policies and vanilla trained deep neural policies.

Table 1: Area under the curve of performance drop under action modification (AM) $a_2$ and $a_w$ for the state-of-the-art adversarially trained deep neural policies and vanilla trained deep neural policies.

| Environments | BankHeist | | RoadRunner | | Freeway | |
|---|---|---|---|---|---|---|
| Training Method | Adversarial | Vanilla | Adversarial | Vanilla | Adversarial | Vanilla |
| AM $a_2$ | 0.449±0.007 | 0.191±0.04 | 0.414±0.015 | 0.247±0.009 | 0.351±0.009 | 0.302±0.007 |
| AM $a_w$ | 0.311± 0.011 | 0.398±0.011 | 0.345±0.011 | 0.393±0.009 | 0.241±0.007 | 0.311±0.010 |

of states. Our aim is to provide an analysis on how accurate the state-action value function is in representing values for both optimal and non-optimal actions for vanilla trained deep neural policies and state-of-the-art adversarially trained deep neural policies.

### 6.1 INACCURACY OF STATE-ACTION VALUES FOR NON-OPTIMAL ACTIONS

In Figure 1 we show the performance drop $\mathcal{P}_2(p)$ as a function of the fraction of states $p$ in which the action modification is applied for state-of-the-art adversarially trained deep neural policies and vanilla trained deep neural policies. In particular, the action modification is set for the second best action $a_2$ decided by the state-action value function $Q(s, a)$. As we increase the fraction of states in which the action modification set to $a_2$ is applied, we observe a performance drop for both of the deep neural policies. However, we observe that the vanilla trained deep neural policies experience a lower performance drop with this modification. Especially in BankHeist we observe that the performance drop does not exceed $0.55$ even when the action modification is applied for a large fraction of the visited states for the vanilla trained deep neural policies. This gap in the performance drop between the adversarially trained and vanilla trained deep neural policies indicates that the state-action value function learnt by vanilla trained deep neural policies has a better estimate for the non-optimal actions. As we measured the impact of $a_2$ modification on the policy performance, we further test $a_w = \arg\min_a Q(s, a)$ modification (i.e. worst possible action in a given state modification) on the deep neural policy. Figure 2 shows that the performance drop $\mathcal{P}_w(p)$ is higher in the vanilla trained deep neural policies compared to adversarially trained deep neural policies when the action modification is set to $a_w$. This again further demonstrates that the state-action value function learnt by the vanilla trained deep neural policy has a more accurate representation over the non-optimal actions. We hypothesize that adversarial training places higher emphasis on ensuring that the highest ranked action (i.e. the action that maximizes the state-action value function in a given state) does not change under small $\ell_p$-norm bounded perturbations, rather than accurately computing the state-action value function as discussed in Section 3. Since historically $Q$-learning suffered from overestimation of $Q$-values, a method which places higher emphasis on the highest ranked action risks converging to a state-action value function with overestimated $Q$-values. We further demonstrate this in Section 6.2.

### 6.2 OVERESTIMATION OF Q-VALUES IN ADVERSARIALLY TRAINED DEEP NEURAL POLICIES

Overestimation of $Q$-values was initially discussed by Thrun & Schwartz (1993) as a byproduct of the use of function approximators, and was subsequently explained as being caused by the use of the max operator in $Q$-learning (van Hasselt, 2010). Furthermore, overestimation bias resulting in learning of sub-optimal policies was demonstrated in practice by Hasselt et al. (2016). In this subsection we empirically demonstrate that state-of-the-art adversarial training indeed leads to overestimation in $Q$-values, as hypothesized in Section 3. In particular, Figure 4 and Table 2 show the overestimation bias on the state-action values learned by the state-of-the-art adversarially trained deep neural policies.

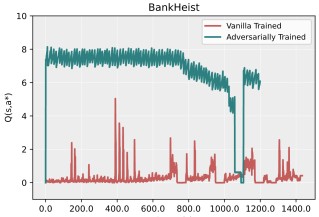 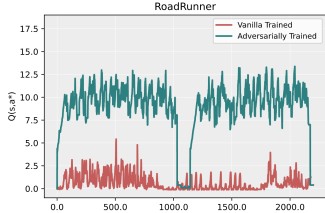 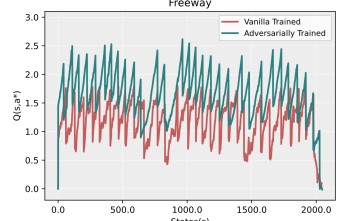

Figure 4: $Q$-value of the best action $a^*$ over the states for the state-of-the-art adversarially trained deep neural policy and vanilla trained deep neural policy.

Table 2: Average $Q$-values of the optimal action in state-of-the-art adversarially trained deep neural policies and vanilla trained deep neural policies.

| Environments | BankHeist | | RoadRunner | | Freeway | |
|---|---|---|---|---|---|---|
| Training Method | Adversarial | Vanilla | Adversarial | Vanilla | Adversarial | Vanilla |
| $Q(s,a^*)$ | 5.903±2.052 | 0.300±0.434 | 8.806±3.216 | 0.602±0.781 | 1.667± 0.406 | 1.185±0.348 |

Considering that overestimation bias is still an issue and active area of research for vanilla deep neural policy training (Lan et al., 2020; Anschel et al., 2017; Kuznetsov et al., 2020), the additional bias introduced intrinsic to adversarial training must be addressed to be able to learn optimal policies.

### 6.3 INCONSISTENCIES IN ACTION RANKING IN ADVERSARIALLY TRAINED DEEP NEURAL POLICIES

In this subsection we demonstrate the inconsistencies in the non-optimal action ranking in adversarially trained policies. In particular, in Figure 3 in BankHeist choosing the worst action leads to a smaller performance drop than choosing the second best action i.e. $\mathcal{P}_w(p) < \mathcal{P}_2(p)$ for all $p$. Thus, this demonstrates that the state-action value function is not ranking the sub-optimal actions accurately. While learning an accurate representation of the state-action values is important for obtaining a policy that aims to maximize expected cumulative rewards, learning the correct order of the actions can also solve this problem. Furthermore, in some cases the deep neural policy indeed must know the correct order

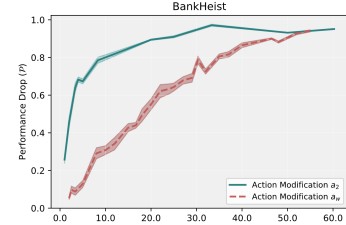

Figure 3: $\mathcal{P}_2$ and $\mathcal{P}_w$ for adversarially trained deep neural policies.

of the actions due to the presence of an obstruction that blocks the optimal action either due to the existence of other agents or environmental effects (Rashid et al., 2020; Gleave et al., 2020). In particular, in safe reinforcement learning several algorithms have been proposed to learn the ranking of the actions so that the agent can choose the next-best ranked action in safety critical situations (Alshiekh et al., 2018). Some work has also pointed out that in some cases learning the relative rank of the actions (Lin & Zhou, 2020) can be more sample efficient than learning correct estimates of the state-action values. While the inconsistency in action ranking for adversarially trained deep neural policies can be seen as a vulnerability problem from a security point of view, most intriguingly these results demonstrate the loss of information in state-action value function as a novel fundamental trade-off intrinsic to adversarial training.

### 6.4 ACTION GAP PHENOMENON

The action gap is defined as the difference between the state-action value of the optimal action and the state-action value of the second ranked action.

$$\kappa(Q,s) = \max_{a' \in \mathcal{A}} Q(s,a') - \max_{a \notin \arg\max_{a' \in \mathcal{A}} Q(s,a')} Q(s,a) \qquad (10)$$

Initially, Farahmand (2011) describes the existence of a large action gap as a desirable property of an MDP, which makes learning an optimal policy easier. Subsequently, Bellemare et al. (2016) proposed a connection between the action gap and the overestimation of $Q$-values, and in particular hypothesized that increasing the action gap of the learned value function causes a decrease in overestimation of $Q$-values. Following this study, several papers built on the hypothesis that increasing

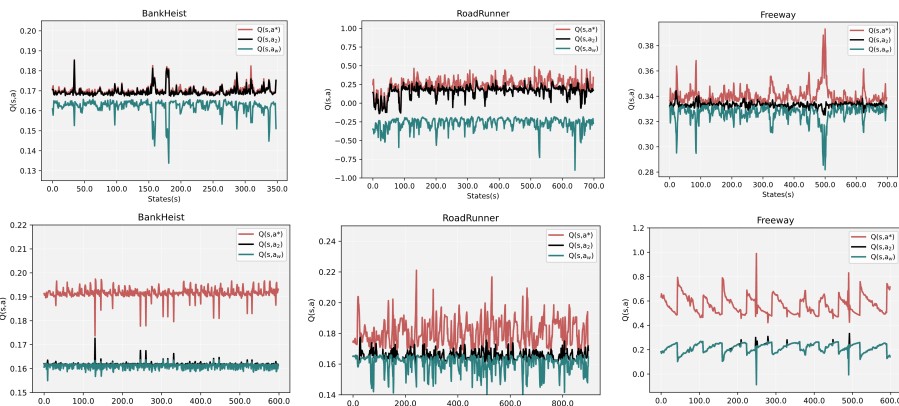

Figure 5: Normalized state-action values for the best action $a^*$, second best action $a_2$ and worst action $a_w$ over states. Row1: Vanilla trained policies. Row2: State-of-the-art adversarially trained policies.

Table 3: Normalized state-action value estimates[1] and state-action value estimate shift for the second best action in state-of-the-art adversarially trained deep neural policies.

| $Q(s,a)$ | $Q(s,a^*)$ | | $Q(s,a_2)$ | | $Q(s,a_w)$ | |
|---|---|---|---|---|---|---|
| ALE | Adversarial | Vanilla | Adversarial | Vanilla | Adversarial | Vanilla |
| BankHeist | 0.1894±0.002 | 0.170±0.003 | 0.130±0.0006 | 0.169±0.002 | 0.127±0.0010 | 0.161±0.004 |
| RoadRunner | 0.1696±0.008 | 0.236±0.094 | 0.132±0.0026 | 0.159±0.079 | 0.126±0.0049 | -0.265±0.071 |
| Freeway | 0.1894±0.002 | 0.341±0.008 | 0.130±0.0006 | 0.333±0.002 | 0.127±0.0010 | 0.325±0.009 |

the action gap causes reduction in bias (Smirnova & Dohmatob, 2020; Fox et al., 2016; Jain et al., 2020; Lu et al., 2019). In Figure 5 we show that adversarial training increases the action gap. Thus, the fact that adversarially trained deep neural policies overestimate the optimal state-action values (see Section 6.2) refutes the hypothesis that increasing the action gap is the sole cause of a decrease in overestimation bias of state-action values. We hypothesize that the consistent Bellman operator (Bellemare et al., 2016) may cause a decrease in overestimation for a different reason. In particular, the consistent Bellman operator corresponds to a special case of a certain reparameterization of Kullback-Leibler regularization for value iteration (Vieillard et al., 2020). Thus, it may be the case that the decrease in overestimation of $Q$-values and improvement in performance is due to a type of implicit regularization rather than to an increase of the action gap. Hence, our results show that increasing the action gap alone may coincide with an increase in overestimation of $Q$-values.

## 7 CONCLUSION

In this paper we focus on the state-action value function learnt via the state-of-the-art adversarially trained deep neural policies and vanilla trained deep neural policies. We provide theoretical analysis on the fundamental effects caused by adversarial training on the state-action value function. Furthermore, we conduct manifold experiments in the Arcade Learning Environment and with our systematic analysis we demonstrate that vanilla trained deep neural policies have more accurate and consistent estimates for the state-action values than the state-of-the-art adversarially trained deep neural policies. More intriguingly, we show that adversarially trained deep neural policies in certain MDPs completely loses all the information in the state-action value function that contains the relative ranking of the actions. More importantly, we show that state-of-the-art adversarially trained deep neural policies learn overestimated state-action values. We believe our investigation lays out intrinsic properties of adversarial training while systematically revealing the underlying vulnerabilities, and can be conducive to building robust and optimal deep neural policies.

---

[1]Note that due to the fact that the adversarially trained deep neural policy overestimates $Q$-values, we introduce a normalization in order to compare the action gaps of adversarially and vanilla trained policies. In particular, in Figure 5 we report normalized $Q$-values in each state $s$ by dividing $Q(s,a)$ by $\sum_a |Q(s,a)|$.

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
