# OpenReview forum: "The Adversarial Regulation of the Temporal Difference Loss Costs More Than Expected"
_ICLR.cc/2023/Conference — Submitted to ICLR 2023_

### Official Review · Reviewer_JcRE · 2022-10-16

**Confidence:** 4
**Correctness:** 3
**Technical Novelty And Significance:** 2
**Empirical Novelty And Significance:** 2
**Recommendation:** 5

**Clarity, Quality, Novelty And Reproducibility:**

CLARITY

The paper could be written more formally and has also some (minor) specific elements that are not accurate, for instance:
- The paper write "Our results essentially demonstrate that vanilla trained deep reinforcement learning policies have more accurate and consistent estimates for the state-action values". I don't believe that the paper actually demonstrates this. It could be rephrased as an empirical evaluation.
- "our investigation lays out intrinsic properties (...) to building robust and optimal deep neural policies.": In deep RL we usually do not expect that the policies will actually be optimal.
- There are many reference to "Example 3", however I do not find any explicit mention of what that example 3 (we can guess that it is the one from the top of page 4)


QUALITY

- Proposition 3.1 and 3.2: There are interesting insights from these propositions but as they are written, it seems that they lack any kind of generality and that they only provide information about a given example. It might be more useful to rewrite them in a more general form (e.g. something along the lines of "if a function approximator of a given type is used, then the regularizer $R(\theta)$ can lead to both overestimation of the first ranked action, and re-ordering of the ranking of the suboptimal actions"). For some propositions, an example could then sufficient as a proof but the propositions would then be more relevant.
- Most experiments are provided with limited relevant baselines and on three environments.
- The appendix mentions a discount factor $\gamma$ of 1. That seems a strange choice as it is known that a discount factor of 1 will in many cases create instabilities and divergence, particularly when used with deep learning as a function approximator and long horizons.

NOVELTY

The paper provides interesting insights for a relevant problem in RL, however the novelty is a bit unclear in terms of contributions. Two key elements that are highlighted in the conclusion are
- "adversarially trained deep neural policies in certain MDPs completely loses all the information in the state-action value function that contains the relative ranking of the actions" and
- "we show that state-of-the-art adversarially trained deep neural policies learn overestimated state-action values"

These are in fact common problems observed for RL with function approximators and it is unclear how this paper provides clear insight on whether this is generally worse/better because of the relatively limited set of experiments and of the relatively narrow theory that is developed.


REPRODUCIBILITY:
The source code is not provided. The papers does provide some of the experiments in the appendix.

**Strength And Weaknesses:**

Strengths:
- This is an important area of study
- The paper highlights some problems that were not fully investigated in the past

Weaknesses
- The contributions are not very strong in the current state of the paper (see below).

**Summary Of The Paper:**

The paper studies a regularization that can improve the learning of Q-values when used in combination with deep learning. The paper shows that these regularization techniques can cause inconsistencies and overestimations in the state-action value functions while vanilla trained deep reinforcement learning policies have more accurate estimates for the Q-values.

**Summary Of The Review:**

The paper looks into an interesting problem and has initial good contributions. There are however elements that need to be improved both on the form and on the substance.

---

> ### Author Response · Authors · 2022-11-13
> **Author Response**
>
> Thank you for providing feedback on our paper. Below we address your questions.
>
> 1. *“ It might be more useful to rewrite them in a more general form “*
>
> Thank you very much for this suggestion. Now we have written in a more general form in Section 3, and updated our paper.
>
> 2. *”These are in fact common problems observed for RL with function approximators ”*
>
> The questions you have asked under the Novelty title make us conjecture that you have perhaps missed some details in the experiments section. First of all, all of the experiments in Section 6 provide comparisons to “RL with function approximators” (i.e. vanilla trained deep reinforcement learning policies). Thus, while Section 6.2 is dedicated to providing comparisons between vanilla trained policies and adversarially trained policies, Table 2 demonstrates that the adversarially trained deep reinforcement learning policies learn approximately 10 times higher state-action values than the baseline RL with function approximators. Hence, these results clearly demonstrate that adversarially trained policies learn overestimated state-action values.  The fact that the certifiable robust policies learn overestimated state-action values across several tasks and across several certifiable adversarial training algorithms,  is indeed one of the novel contributions of our paper. This was clearly not known prior to our paper. Can you please specifically claim what you exactly think is not novel with this contribution?
>
> The fact that our paper provides comparisons directly to the **vanilla trained policies** (i.e. reinforcement learning with function approximators), and demonstrates that indeed vanilla trained policies **do not lose** any information in the state-action value function that contains the relative ranking of the actions, clearly shows that this is not a common problem observed for RL with function approximators as you claim.
>
> 3. *”Relevant baselines”*
>
> Please note that we use the **exact same MDPs** with a similar number of games **from the original studies** that proposed these adversarial training techniques [1,2].
>
> [1] Robust Deep Reinforcement Learning against Adversarial Perturbations on State Observations, NeurIPS 2020. [Spotlight Presentation]
>
> [2] Robust Deep Reinforcement Learning through Adversarial Loss, NeurIPS 2021.
>
> 4. *”Appendix”*
>
> Sorry for this. This is clearly a typo in the appendix. The value of $\gamma$ is 0.99.

---

> > ### Comment · Reviewer_JcRE · 2022-11-14
> > **Additional comments**
> >
> > Thanks for the clarification (comment 3) and the adaptations (comments 1 and 4).
> >
> > To write in a slightly more general form the propositions (following comment 1), it seems that the authors have introduced a number of elements that do not seem correct or are introduced in an unclear way. Here are three elements:
> > - Beginning of page 4: "$1 > \lambda > 0$ be the discount factor, and let $\delta > \eta > 0$ be small constants with $\lambda > \delta > \eta$": Why is $\lambda$ used as discount factor and not $\gamma$? (Why is $\delta > \eta$ written twice?)
> > - Why is the following expression = to 1? $Q^∗(s1, a1) = Q^∗(s2, a2) = \sum_{t=0}^\infty (1 − λ)λ^t = 1$
> > - Why do you make the "state vector" satisfy some equalities that are functions of $\theta_i^*$? $\theta_i^*$ are supposed to be parameters that are optimized to fit the Q-values?
> >
> > Comment 2 remains a main concern from my review. From the theory perspective the elements presented in the paper show in Theorem 3.3 (assuming the derivations are correct, which I'm unsure of at this point, see above) that "the regularizer $R(\theta)$ can lead to overestimation of the value of the optimal action, and re-ordering of the values of the suboptimal actions". However, this is also true without any regularizer for Q-learning with function approximators.

---

> > > ### Author Response · Authors · 2022-11-15
> > > **Response**
> > >
> > > Thank you for your comments.
> > >
> > > 1. *“$\gamma$ and $\lambda$”*
> > >
> > > We have now switched to using $\gamma$ for the discount factor and fixed the repetitiveness in the definition of $\delta$ and $\eta$.
> > >
> > > 2. *”Why is the following expression = to 1?”*
> > >
> > > Note that for $|\gamma| < 1$ the geometric series converges $\sum_{t=0}^{\infty} \gamma^t = \frac{1}{1-\gamma}$. Hence, $\sum_{t=0}^{\infty} (1-\gamma)\gamma^t = (1-\gamma)\cdot \frac{1}{1-\gamma} = 1$.
> > >
> > > 3. *”State feature vectors”*
> > >
> > > We also clarified the definition of the state feature vectors by writing them in terms of three orthonormal vectors $z_1,z_2,z_3$ and then concluding that the optimal linear state-action value function has parameters $\theta^* = (\theta_1^*,\theta_2^*,\theta_3^*)$ where $\theta_i^* = z_i$.
> > >
> > > 4. *“ Comment 2 concern”*
> > >
> > > The problems like overestimation are indeed lasting problems we deal with in reinforcement learning with function approximators. The initial reason that double $Q$-estimators were introduced [1] for instance is to target this problem [2]. In our paper, in Section 6.2 we discuss how the overestimation problem is an issue in reinforcement learning, and why it is important. However, in our paper we show that on top of function approximators in reinforcement learning the recent certifiable adversarial training techniques **independently** introduce **further bias** into the deep neural policy. Furthermore, these state-of-the-art adversarial training techniques that are aimed to make the policies robust degrade the quality of the state-action value function
> > >
> > > Theorem 3.3. demonstrates that even when the function approximation error is zero, the regularizer will introduce an independent source of error to the reinforcement learning policy.
> > > In particular, the adversarial training introduces an **independent source of error** to the state-action value function **on top of the function approximation error**. See that currently there is a strong effort on proposing various certified adversarial training techniques in deep reinforcement learning to make these policies robust [3,4]. However, the effects of these training techniques have not been discussed. These training techniques [3,4] while claiming to make policies robust, bring the lasting problems in reinforcement learning that the community has put extensive efforts to resolve over the decades [1,2,3] back to the surface in a more severe way.
> > >
> > >
> > > [1] Hado van Hasselt and Arthur Guez and David Silver. Deep Reinforcement Learning with Double Q-learning, AAAI 2016.
> > >
> > > [2] Hado van Hasselt. Double Q-learning. NIPS 2010.
> > >
> > > [3] Sebastian Thrun and Anton Schwartz. Issues in using function approximation for reinforcement learning. 1993.
> > >
> > > [4] Robust Deep Reinforcement Learning against Adversarial Perturbations on State Observations, NeurIPS 2020. [Spotlight Presentation]
> > >
> > > [5] Robust Deep Reinforcement Learning through Adversarial Loss, NeurIPS 2021.

---

> ### Author Response · Authors · 2022-11-25
> **Gentle Reminder**
>
> Dear Reviewer  JcRE,
>
> We would like to thank you again for the time you have invested in providing feedback for our paper. We would highly appreciate it if you could confirm that our response addresses your questions. We would also be able to engage further if you had additional questions that you would like to discuss.
>
> Best regards,
>
> Authors

---

### Official Review · Reviewer_AQqn · 2022-10-24

**Confidence:** 4
**Correctness:** 2
**Technical Novelty And Significance:** 2
**Empirical Novelty And Significance:** 2
**Recommendation:** 3

**Clarity, Quality, Novelty And Reproducibility:**

Good clarity. The novelty is limited. Code is not provided and implementation details are missed. Hard for reproducibility.

**Strength And Weaknesses:**

Strengths:
1. The observation, that vanilla trained deep neural policies have more accurate and consistent estimates for the state-action values than the state-of-the-art adversarially trained deep neural policies, is counterfactual and interesting.

Weaknesses:
1. The theoretical analysis doesn’t support much about the empirical study. The analysis assumed that action-value function is linear and as simple as an inner product, which can not be extended to the cases of empirical studies.
2. Lack of contribution. Demonstrating the problem without providing potential solutions may not quite yet reach the high bar set for ICLR.
3. The main claim needs more investigation or explanations.


**Summary Of The Paper:**

This paper investigates that adversarially trained deep neural policies can potentially lead to inconsistencies and overestimations in the state-action value functions. A linear case theoretical analysis and empirical studies on Arcade Learning Environment are provided to verify the observation.

**Summary Of The Review:**

Overall, the depth of this contribution may not quite yet reach the high bar set for ICLR. The main claim, that vanilla trained deep neural policies have more accurate and consistent estimates for the state-action values than the state-of-the-art adversarially trained deep neural policies, needs more investigations.

The paper defines a performance drop metric as a measure of the accuracy of state-action values.
The authors pick a random state $s$ during a rollout, make the agent choose action $a_i$ in state $s$, and in all other states have the agent choose the best Q-value actions. The performance drop due to the action modification is measured. The authors claim that adversarially training would make the performance drop large and hence is not accurate.

However, all the experiments are done on discrete action spaces, and I would expect there would be a big change in performance due to action modifications, for example, turning right in a maze changed to turning left would impact a lot for finding the exit. I think the performance drop due to action modification could be reasonably large and large performance drop actually shows that the Q-value function could discriminate good actions from bad actions well, which is the target of adversarial learning.

I also kind of disagree with the overestimation claim. Table 2 doesn’t show the true average Q-values, so that we don’t know which one is overestimated. To me, the value scale (<10) is still under control. The target of Q-value function is to help the learning of policies, so as long as the good action regions are assigned higher and reasonable Q values, Q function could provide good learning signals for the policy.

Implementation details are not discussed, and the work is not reproducible. The paper only mentions using state-of-the-art adversarially trained deep neural policies, while not includes the actually used algorithm.

---

> ### Author Response · Authors · 2022-11-11
> **Author Response**
>
> Thank you for the time you have allocated in providing feedback.
>
>
> 1. *”Large performance drop actually shows that the Q-value function could discriminate good actions from bad actions well.”*
>
> You have a **significant confusion** here. The performance drop $\mathcal{P}_2$ obtained by the action modification $a_2$ and the performance drop $\mathcal{P}_w$ obtained by the worst action modification $a_w$ refer to different things. The relative order of  $\mathcal{P}_2$ and  $\mathcal{P}_w$ is completely switched for adversarially trained policies (see Figure 1, Figure 2 and Table 1 in the paper). Thus, this means the values that adversarially trained policies assign to the state-action value estimates of the suboptimal actions are completely **wrong**.
>
>
> 2. *"The authors claim that adversarially training would make the performance drop large and hence is not accurate."*
>
> Again you have some **confusion** here. From your comment it seems like you think the inaccuracy comes from a large performance drop. This is **false**. The relative order of  $\mathcal{P}_2$ and $\mathcal{P}_w$ demonstrates the inaccuracy in the state-action values not the absolute values of $\mathcal{P}_w$ and $\mathcal{P}_2$. In particular,  Figure 1 reports the results for $\mathcal{P}_2$ for both adversarially trained and vanilla trained policies. Here the performance drop $\mathcal{P}_2$ for adversarially trained policies is **higher** than vanilla trained policies. On the other hand, Figure 2 reports results $\mathcal{P}_w$ for both adversarially trained and vanilla trained policies.  Here the performance drop $\mathcal{P}_w$ for adversarially trained policies is **lower** than vanilla trained policies. Hence, once more the inaccuracy of the state-action value function comes from the relationship between $\mathcal{P}_w$ and $\mathcal{P}_2$.
>
> 3. *”While not includes the actually used algorithm”*
>
> This is already included in the paper in many different Sections multiple times. In particular, this information is already present in the paper in Section 2.3, in Section 3 and in Section 5.
>
> 4. *”To me, the value scale (<10) is still under control.”*
>
> Can you clarify based on what you think **more than 20 times higher** Q values for adversarially trained policies seems okay to you?
>
> 5. *“Implementation details are not discussed*”
>
> Implementation details are indeed discussed in the supplementary material.
>
> 6. *”Table 2 doesn’t show the true average Q-values”*
>
> Since this is deep reinforcement learning, you will not know what the true Q-values are. But what you can know is the true average returns. Below you can find the true average returns reported in the Table. The fact that vanilla trained policies have approximately similar but slightly higher scores and the adversarially trained policies assign higher state-action values, once more demonstrates that adversarially trained policies overestimate.
>
> | MDPs                     | Adversarially Trained             |  Vanilla Trained             |
> |---------------------------|  :---------------------------: |   :--------------------------------: |
> | BankHeist          |  $1228.7\pm 11.4$                 |   $1302.9  \pm 22.8 $      |
> | RoadRunner      |   $43190.0 \pm 8154.0   $       |    $46271.0 \pm 7182.0$ |
> | Freeway             |   $30\pm 0.0   $                      |    $32.0 \pm 0.3   $          |
>
>
> 7. “*The theoretical analysis”*
>
> The theoretical motivation provided in Section 3 is dedicated to understand and explain the empirical results obtained in Section 6. Hence, the theoretical analysis in Section 3 indeed supports the empirical study and provides insights into why we observe these results empirically.
>
> 8. *“Demonstrating the problem without providing potential solutions may not quite yet reach the high bar set for ICLR.”*
>
> We believe this statement might be considered as simply **false**. Please see the field of adversarial machine learning [1,2,3,4,5], in particular the line of research that solely focuses on explaining and understanding the vulnerabilities of the machine learning models. The adversarial machine learning field has two parallel research directions that target, as a future research goal, to help in building robust models: while one focuses on demonstrating and explaining the vulnerabilities [1,2,3,4,5], the other one focuses on methods to evade the vulnerabilities [6]. Nonetheless, these two lines of research jointly help the field move forward.
>
> [1]  Poisoning and Backdooring Contrastive Learning, ICLR 2022.
>
> [2] On Adaptive Attacks to Adversarial Example Defenses, NeurIPS 2020.
>
> [3] Deep Reinforcement Learning Policies Learn Shared Adversarial Features Across MDPs, AAAI 2022.
>
> [4] Label-Only Membership Inference Attacks, ICML 2021.
>
> [5] Stealthy and Efficient Adversarial Attacks against Deep Reinforcement Learning, AAAI 2020.
>
> [6] Robust Deep Reinforcement Learning against Adversarial Perturbations on State Observations, NeurIPS 2020.

---

> > ### Comment · Reviewer_AQqn · 2022-11-16
> > **Re: Response**
> >
> > Thanks for providing the response. The reponse does not convince me and hence I keep my evaluation for this work.
> >
> > * I now understand your point that the inaccuracy of state-action value estimates from adversarial traning is identified by the re-ordering of the values of the suboptimal actions (seen by $a_w>a_2$), though the re-ordering discussion is not provided in Sec 6.1 "INACCURACY OF STATE-ACTION VALUES FOR NON-OPTIMAL ACTIONS".
> >
> >   Moreover, I don't think the reordering of suboptimal actions is a disadvantage of adversarial training. With the adversarial regularizer applied, the objective function referenced in Eq. (4) encourages the Q to assign high values for optimal actions while low values for the others. This will benefit the policy to discriminate good actions from bad actions. The value estimates of suboptimal actions could be modified due to the regularization. This is the goal of adversarial learning.
> >
> > * Spliting the reviewer's question and answering to each part is not a good way for communication. My second concern is the comparison that you made in Table 2 are not meaningful or fair, because you don't know the good Q-values here. Why the low Q values could be better? Why the higher Q values are considered to be overestimated? The target of Q is for guiding the policy to learn good actions. For example, with the Q network fixed, you multiply you all Q values by 20, that won't change much about the policy learning but only the learning rate. For another example, in continuous task setting. for HalfCheetah-v3, your Q-values could reach 1000, but it still works good for policy learning.
> >
> >   Your return results actually show that the Q-value estimates from adversarial learning are good here. The performance from the both two methods are comparable.
> >
> > * Third concer is still there. The theoretical analysis assumed that action-value function is linear and as simple as an inner product, which can not be extended to the cases of empirical studies, where non-linear neural networks are applied.

---

> > > ### Author Response · Authors · 2022-11-16
> > > **Response**
> > >
> > > Thank you for responding.
> > >
> > > 1. *“Reordering of suboptimal actions”*
> > >
> > > Please note that certifiable adversarial training techniques are proposed to build **robust** deep reinforcement learning policies. However, from the security point of view losing the information on the suboptimal actions in fact makes these certifiable adversarial training techniques **extremely vulnerable** to action attacks. Thus, losing this information on the state-action value function accuracy does indeed come with a cost, and hence is a disadvantage.
> > >
> > >
> > > 2. *”Overestimation“*
> > >
> > > Can you please see Figure 3 in Page 5 of [1]? As you can see here the $Q$ values that have been discussed to be overestimated by DQN [2] are 15 for Alien, 9 for Space Invaders, 2 for TimePilot, 8 for Zaxxon. Our paper is in the Arcade Learning Environment just like [1,2] not in Mujoco in the example you give. Thus, on the contrary to what you are claiming Table 2 **indeed** demonstrates the **overestimation of $Q$ values for adversarially trained policies**.
> > >
> > >
> > > [1] Hado van Hasselt and Arthur Guez and David Silver. Deep Reinforcement Learning with Double Q-learning. AAAI 2016.
> > >
> > > [2] Human-level control through deep reinforcement learning, Nature 2015.
> > >
> > >
> > > 3. *”Theoretical Analysis”*
> > >
> > > The theoretical analysis is based on reinforcement learning with linear function approximation. In general, there are no known theoretical results for analyzing the convergence of deep reinforcement learning algorithms (i.e. with deep neural network function approximation), and hence it is **standard** in reinforcement learning research to use the simpler case of linear function approximation to gain theoretical motivation and mathematical intuition for the behavior of these algorithms. Reinforcement learning with linear function approximation represents one of the most general classes of function approximators for which theoretical results are known, thus this is the reason why the theoretical analysis is conducted with linear function approximation. Furthermore, the theoretical analysis for linear function approximation aligns closely with the empirical results observed with deep neural network function approximation, demonstrating that the model of linear function approximation does indeed give accurate intuitions for the more general neural network setting.

---

### Official Review · Reviewer_2Ms4 · 2022-10-25

**Confidence:** 4
**Correctness:** 2
**Technical Novelty And Significance:** 2
**Empirical Novelty And Significance:** 2
**Recommendation:** 5

**Clarity, Quality, Novelty And Reproducibility:**

The main message of the paper is clearly conveyed, but many important experiment details are not clear and I do not believe I could reproduce the experimental results based on the description in the appendix. Further, the description of the adversarial regularizer used in both the theoretical analysis and empirical investigation differs in an important way from that used by the main baseline SA-DQN as it is not lower bounded. This is an important detail and I worry that many of the results may be an artifact of this distinction, and that the effect sizes seen in the paper would be much smaller under a bounded adversarial regularizer.

I haven’t previously seen an analysis of the effect of adversarial regularization on the action ranking of the Q-function in RL, though the more general robustness of value-based deep RL agents’ action rankings to optimization noise has been studied previously.

**Strength And Weaknesses:**

Strengths:

- The paper offers some interesting insights on the trade-offs between adversarial robustness and accuracy of the action-value function in value-based deep RL.
- The toy example in Proposition 3.2 is a nice illustration of the instability of action rankings in adversarially robust agents.
- The discussion of the adversarial robustness literature seems reasonably comprehensive (though this isn’t my primary field).
- The metric used to estimate action-value accuracy is a smart way of trying to marginalize out noise in the agent trajectories in order to compare the action rankings.
- The finding of section 3 is somewhat surprising — I would not have anticipated such a large gap between the actions of the second-best and worst predicted action.


Weaknesses

- While the literature review discusses the adversarial RL literature, it does not cover many recent works studying the robustness of the action-value ranking of deep RL agents to non-adversarial perturbations. In particular, prior work [1] has shown that even gradient steps computed on disjoint subsets of the replay buffer can change the agent’s ranking over actions in a surprisingly large fraction of the state space. In the context of this prior work, the observation that the ranking of sub-optimal actions changes under adversarial regularization is unsurprising.
- It is widely observed that DQN outputs tend to systematically underestimate the true value of a state. As a result, it seems to me difficult to say whether the values output by the adversarially trained networks are indeed over-estimations of the true value, or whether they are simply accurate estimations of the true action-value function. I am therefore unconvinced as to whether the subsection’s title is necessarily accurate, and would like to see a comparison between the action-values and the true discounted return obtained by the policy to be convinced of this.
- The description of the adversarial regularizer is inconsistent with that described by Zhang et al. in their presentation of SA-DQN. Zhang et al. include a clipping hyperparameter $c$ which lower bounds the adversarial action gap and avoids the pathology of the value estimate of the optimal action growing arbitrarily large. I am curious about the value of $c$ used in this paper.
- Similarly, the value of the regularizer coefficient $\kappa$ is only briefly mentioned in the final  sentence of the supplementary material, and it is said to be in a set of values. However, I could not determine which value was used for the paper’s experiments, nor could I find any results describing the relationship between the phenomena described in the main body and the values of $\kappa$ and $c$. The epsilon value used for epsilon-greedy exploration is quite low (0.02). These values will likely play a large role in determining effect size of the phenomena described in this paper.

**Summary Of The Paper:**

This paper studies the effect of adversarial training on the action-value estimates of value-based deep RL agents. The paper considers both a toy learning problem where the optimal function approximator with and without adversarial regularization can be computed analytically, as well as networks trained on the arcade learning environment. It shows that adversarial training reduces the accuracy of the network’s ranking over actions at a given environment state, that it increases the gap between the action with the highest predicted value and the value of the second-best action, and that it increases the value of the action with the highest predicted value.


**Summary Of The Review:**

This paper highlights an important tradeoff in adversarial training of value-based deep RL agents: that optimizing for the robustness of the greedy policy may have unintended effects on the sub-optimal actions. The paper suffers from two principal weaknesses: firstly, that prior work has already identified that the relative ranking of actions in DQN-like agents is unstable, and so the findings here seem to be a natural consequence of that observation coupled with the focus of adversarial regularization on only the greedy action. Secondly, the regularization scheme under consideration differs from that studied in prior work in a way that would exacerbate the effects observed in the paper, and lack of detail on the experiment setup description makes it difficult to identify whether the experimental findings would reflect practical implementations of state-of-the-art adversarial training schemes for RL.

---

> ### Author Response · Authors · 2022-11-11
> **Clarification on Reference**
>
> Would it be possible for you to write what you mean by [1]? Currently there is no reference for [1] in your review.

---

> > ### Comment · Reviewer_2Ms4 · 2022-11-15
> > **Reference**
> >
> > Apologies for the omission, the reference [1] refers to "The Phenomenon of Policy Churn" (https://arxiv.org/abs/2206.00730).

---

> > > ### Author Response · Authors · 2022-11-15
> > > **Response**
> > >
> > > Thank you for this reference.
> > >
> > > The changes in action ranking during training time at each gradient step with **disjoint subsets** of the replay buffer under policy churn is a **completely different phenomenon** than the one we describe in our paper throughout the experiments. Our paper focuses on the actual expected cumulative rewards achieved when switching to non-optimal actions at **test time** (i.e. once the training is completed).
> > >
> > > To understand this more clearly, observe that due to policy churn, two training runs with disjoint subsets of the replay buffer might encounter that the greedy policy is changing during training. However, if we were to run our experiments on these two policies produced, switching to the second ranked action $a_2$ will yield a smaller performance drop $\mathcal{P}_2$ than switching to the lowest ranked action $a_w$ for each policy respectively (see Figure 1 and Figure 2 that how the vanilla trained policies have **consistent** state-action value functions across MDPs). Thus, even though the policy experiences changes in action ranking during training, once the training is over the relative performance ranking between higher and lower ranked actions is **consistent**.
> > >
> > > Note that  in our paper the corresponding meanings of the actions are tightly connected to the actual cumulative discounted rewards obtained by the agent. Thus, Figure 1 and Figure 2 report results for $\mathcal{P}_2$ and $\mathcal{P}_w$ for adversarially trained policies and vanilla trained policies. In particular, $a_w$ modification results in **lower** expected cumulative rewards than $a_2$ modification for vanilla trained policies, while for adversarially trained $a_w$ modification results in **higher** expected cumulative rewards than $a_2$ modification. These results once more demonstrate that while vanilla training learn policies that contain the correct ordering of the actions that results in corresponding expected cumulative rewards obtained, adversarially trained policies learn **inaccurate** and **inconsistent** state-action value functions.

---

> > > > ### Comment · Reviewer_2Ms4 · 2022-11-27
> > > > **Clarifying connection**
> > > >
> > > > The authors are correct that the higher-level phenomenon studied in their paper is distinct from that of policy churn. However, the  observation that the relative rankings of actions in policy-based agents are easily changed during optimization seems to hold explanatory power in the adversarial training setting as well. In particular, when the optimization objective does not prioritize the accuracy of action rankings (as will be the case for sub-optimal actions under adversarial training), this accuracy likely to decline over the course of training.

---

> ### Author Response · Authors · 2022-11-11
> **Author Response**
>
> Thank you for providing feedback on our paper. Below we address your questions.
>
> 1. *”It is widely observed that DQN outputs tend to systematically underestimate the true value of a state”*
>
> We would like to highlight that this statement is a **misapprehension**. On the contrary, it is widely known that DQN **overestimates** [1]. This overestimation is even the intrinsic reason why Double-$Q$ learning has been invented [2].
>
> [1] Hado van Hasselt and Arthur Guez and David Silver. Deep Reinforcement Learning with Double Q-learning, AAAI 2016.
>
> [2] Hado van Hasselt. Double Q-learning. NIPS 2010.
>
>
> 2. *”The true discounted return obtained by the policy”*
>
> Below we attached the table that reports the average return obtained by these policies. The true discounted return obtained by the vanilla trained and adversarially trained policy while being approximately similar, for all of the MDPs the vanilla trained policies has a higher score (i.e. true return). Thus, the fact that adversarially trained policies have higher state-action value estimates once more demonstrate that adversarially trained policies learn overestimated state-action values.
>
>
> | MDPs                     | Adversarially Trained             |  Vanilla Trained             |
> |---------------------------|  :---------------------------: |   :--------------------------------: |
> | BankHeist          |  $1228.7\pm 11.4$                 |   $1302.9  \pm 22.8 $      |
> | RoadRunner      |   $43190.0 \pm 8154.0   $       |    $46271.0 \pm 7182.0$ |
> | Freeway             |   $30\pm 0.0   $                      |    $32.0 \pm 0.3   $          |
>
> 3. *”$ c$ and $\kappa$”*
>
> Our implementation of the adversarial regularizer is the exact same implementation as the SA-DDQN paper. Thank you very much for pointing out the typo here. Indeed there is a $c$ in the regularizer.
> The value of $c$ is 1, which is exactly the same as the reference implementation of the SA-DDQN paper.
> Similarly, the value of $\kappa$ used in the experiments is 0.005, exactly as in the reference implementation from the original paper.
> We kept these hyperparameters exactly the same as the SA-DDQN implementation in order to demonstrate that the issues we describe arise for the SA-DDQN algorithm precisely as it was originally implemented.
>
> Please also further note that the phenomenon described in our paper extends beyond  SA-DDQN too. There are more results in the supplementary material that shows that the issues described in SA-DDQN also exist and persist in a more recent adversarial training technique. Thus, demonstrating the loss of information in the state-action value functions as a novel fundamental trade-off intrinsic to adversarial training.
>
>
> Thank you again for the time you have invested in the review. We hope that our response addresses your questions.

---

> > ### Comment · Reviewer_2Ms4 · 2022-11-27
> > **Clarification**
> >
> > Thanks to the authors for this response. I appreciate the clarification on the SA-DDQN implementation; this has resolved one significant concern I'd had about the paper.
> >
> > Regarding the over-estimation of Q-values, it is true that in the large training time limit and particularly in tabular environments "it is widely known that DQN overestimates [1]. This overestimation is even the intrinsic reason why Double- learning has been invented [2]." However, I and many others I've spoken to who have trained Q-learning agents with function approximation have observed that the learned Q-values tend to under-estimate the return of the policy for a long time during training. This is particularly evident in settings such as Atari, where the agent may score hundreds of thousands of points but have an estimated Q-value in the single-digits, but it also can be observed even in Cartpole: the agent under-estimates the value for a long time after it has learned an optimal policy, as it takes time for the TD backups to propagate the magnitude of the value function through the state space.

---

> > > ### Author Response · Authors · 2022-11-27
> > > **Response**
> > >
> > > Thank you very much for your response and for your appreciation for the clarifications. One important thing that needs to be further clarified here is that everything reported in our paper is during **test time** (i.e. as it is referred to in adversarial deep reinforcement learning, **once the training is completed**). Thus, underestimation that occurs during early training time is not relevant due to the fact that Table 2 reports state-action values once the training is completed.
> > >
> > > Please see Figure 3 in Page 5 of the paper [1]. Another important thing to note here is that Figure 3 of paper [1] reports the value estimates of DQN which are notably overestimated. Please see that the scale of the overestimated state values of DQN for the Atari environment is reported to be between 2 and 2.5 for TimePilot, 6 and 8 for Zaxoon, 8 and 10 for Space Invaders, and 15 and 20 for Alien.
> > >
> > > [1] Hado van Hasselt and Arthur Guez and David Silver. Deep Reinforcement Learning with Double Q-learning. AAAI 2016. [[Paper Link]](https://arxiv.org/pdf/1509.06461.pdf)
> > >
> > >
> > > We highly appreciate the effort you have put in providing feedback. We hope that our clarifications addressed your questions.

---

> ### Author Response · Authors · 2022-11-17
> **Kind Reminder**
>
> The author response period will come to an end tomorrow. Would it be possible for you to let us know if your concerns have been addressed? We are also happy to discuss if you have any further questions.

---

> ### Author Response · Authors · 2022-11-29
> **In Light of our Clarifications**
>
> Dear Reviewer 2Ms4,
>
> Thank you very much again for your review and response. If our response addressed your questions would it be possible for you to reconsider the score you have assigned to our paper in light of our clarifications?
>
> Best regards,
>
> Authors

---

### Decision · Program_Chairs · 2023-01-20

**Decision:**

Reject

**Justification For Why Not Higher Score:**

A paper identifying a set of predictable effects of an existing method is often expected to at least offer some practical insight into whether, and if so how, to avoid or mitigate these effects. In its current form, the paper doesn't offer much new insight from this perspective.

**Justification For Why Not Lower Score:**

N/A

**Metareview: Summary, Strengths And Weaknesses:**

The initial reviews questioned whether some specific claims made by the authors are correct/defendable. While the authors' responses have successfully addressed some of these key concerns, in particular the validity of the paper's methodology, concerns about the strength of the paper's contribution remain. The AC would like to rephrase a reviewer's feedback during discussion, which the AC agrees with:

"The paper has three main contributions: an analysis of adversarial regularization in an example linear function approximator, empirical evidence that the predicted value of the greedy action increases in practical adversarial training schemes, and empirical evidence that the value function's ranking over non-greedy actions decreases in accuracy.

1. The linear case study is a fairly straightforward computation and doesn't stand on its own as a significant contribution; the example MDP is somewhat contrived and it is difficult to see how the intuition developed here might generalize to more realistic function approximators.

2. The empirical observation that adversarial training reduces the accuracy of an agent's ranking of sub-optimal actions is a natural implication of prior work on the instability of action-value rankings in value-based RL algorithms.

3. Finally, independent of whether the increase in the predicted Q-value pushes it higher than the true expected return (as has been debated at length in the discussion), it is difficult to see the value of this observation. Increasing the value of the greedy action is a natural way to decrease the sensitivity of the resulting policy to perturbations of its parameters-- it would be surprising if the network didn't increase its predicted value."